# Promising Markers of Inflammatory and Gut Dysbiosis in Patients with Post-COVID-19 Syndrome

**DOI:** 10.3390/jpm13060971

**Published:** 2023-06-08

**Authors:** Ekaterina Sorokina, Alisa Pautova, Oleg Fatuev, Vladislav Zakharchenko, Alexander Onufrievich, Andrey Grechko, Natalia Beloborodova, Ekaterina Chernevskaya

**Affiliations:** 1Federal Research and Clinical Center of Intensive Care Medicine and Rehabilitology, 25-2 Petrovka Str., 107031 Moscow, Russia; sorokina1901@mail.ru (E.S.); alicepau@mail.ru (A.P.); nvbeloborodova@yandex.ru (N.B.); 2Institute of Biochemical Technology and Nanotechnology, Peoples’ Friendship University of Russia, 6 Miklukho-Maklaya Str., 117198 Moscow, Russia; 3Federal State Budgetary Institution “N.N. Burdenko Main Military Clinical Hospital” of the Ministry of Defense of the Russian Federation, Hospital Sq., Build. 3, 105094 Moscow, Russia

**Keywords:** SARS-CoV-2 infection, post-acute COVID-19, long COVID-19, gut microbiota, dysbiosis, microbial metabolites, interleukin-6, succinic acid, fumaric acid, 4-hydroxybenzoic acid

## Abstract

Post-COVID-19 syndrome is a complex of different symptoms, which results in a multisystemic impairment after the suffering from COVID-19 infection. The aim of the study was to reveal the clinical, laboratory, and gut disorders in patients with post-COVID-19 syndrome (*n* = 39) before and after taking part in the 14-day complex program of rehabilitation. A complete blood count, coagulation test, blood chemistry, biomarkers, and metabolites in serum samples, and gut dysbiosis were revealed in patients on the day of admission and after 14-day rehabilitation, in comparison with the variables of healthy volunteers (*n* = 48) or with reference ranges. On the day of discharge, patients noted an improvement in respiratory function, general well-being, and mood. At the same time, the levels of some metabolic (4-hydroxybenzoic, succinic, fumaric acids) and inflammatory (interleukin-6) variables, which were increased on admission, did not reach the level of healthy people during the rehabilitation program. Taxonomy disbalance was observed in patients’ feces, namely, a high level of total bacterial mass, a decrease in the number of *Lactobacillus* spp., and an increase in pro-inflammatory microorganisms. The authors suggest that the post-COVID-19 rehabilitation program should be personalized, considering the patient’s state together with not only the baseline levels of biomarkers, but also with the individual taxonomy of the gut microbiota.

## 1. Introduction

Many people who have suffered from SARS-CoV-2 continue experiencing a range of non-specific symptoms that they did not observe prior to COVID-19 infection for up to 2–6 months. The most frequent symptoms are fatigue, headache, dyspnea, myalgia, hair loss, and some other symptoms, which are called post-COVID-19 syndrome, long COVID, or post-acute COVID. Despite that many people with post-COVID-19 syndrome may have different comorbidities including cardiovascular, metabolic, and other chronic disorders, it is important to reveal the biomarkers that could be specific for post-COVID-19 syndrome [1].

Different original studies [2,3,4,5], meta-analyses [6], and reviews [7,8] focused on the search for the post-COVID-19 syndrome biomarkers and most of them were divided into inflammatory and vascular biomarkers. Some other studies revealed different metabolites and lipoproteins specific for post-COVID-19 patients compared to those in acute phase of disease or healthy people [9,10].

The gut microbiota is known to influence the human health by playing an important role in metabolic, regulatory, pro- and anti-inflammatory, and other processes in the body. The overprescribing of antibiotics in acute COVID-19 treatment inevitably led to an alteration of the gut microbiota composition. The latter together with residual inflammatory processes may be the pathogenetic factors of post-COVID-19 syndrome and should be simultaneously monitored in post-COVID-19 patients [11]. The aim of our study was to reveal the clinical, biochemical, metabolic, and/or gut disturbances in patients with post-COVID-19 syndrome before and after receiving the 14-day complex program of the rehabilitation.

## 2. Materials and Methods

### 2.1. Study Design

The present study was a single-center clinical study, performed at the Federal Research and Clinical Center of Intensive Care Medicine and Rehabilitology (Moscow, Russia). The study was conducted in accordance with the Declaration of Helsinki and approved by the Ethics Committee of Federal Research and Clinical Center of Intensive Care Medicine and Rehabilitology (protocol code PP #4/20 from 22 September 2020). Informed consent was obtained from all patients involved in the study.

For patients with post-COVID-19 syndrome (*n* = 39), the median age was 56 (49–71) years, 12 (31%) were men, they were admitted at the Federal Research and Clinical Center of Intensive Care Medicine and Rehabilitology, and they were enrolled in a complex program of rehabilitation lasting at least7 days. Some patients (*n* = 28), with a median age of 59 (51–66) years, of which 8 (28%) were men, completed the 14-day rehabilitation program. Other patients (*n* = 11) were discharged before the end of the program.

The inclusion criteria: post-COVID-19 patients discharged from the hospital with moderate severity of acute COVID-19 more than 3 months ago; pulmonary foci of consolidation and fibrosis on chest CT (CT 1–4) at acute COVID-19; negative polymerase chain reaction (PCR) result for SARS-CoV-2 on the day of admission to the rehabilitation center; fatigue at a level not seen before COVID-19 on the day of admission to the rehabilitation center.

The exclusion criteria: positive PCR result for SARS-CoV-2; body temperature above 38 °C; respiratory rate over 30/min; increase in systolic blood pressure above 180 mmHg or a decrease below 90 mmHg.

Information about the severity of COVID-19, antibacterial therapy, complications, and the clinical signs of post-COVID-19 syndrome were recorded at the initial examination by the therapist and then retrospectively analyzed from medical documentation.

### 2.2. Post-COVID-19 Rehabilitation Complex

Post-COVID-19 rehabilitation at the Federal Research and Clinical Center of Intensive Care Medicine and Rehabilitology (Moscow, Russia) is aimed at restoration of the respiratory system using a respiratory gymnastics complex with more than 10 different exercises according to the Strelnikova technique, which is based on active inhalation and passive exhalation [12]; and also, physical and motor activity (Terrenkur [13]) with clinical control of the patient’s condition. Special diets are prescribed to all patients according to their comorbidity. Pevsner diets are usually prescribed to most patients. This is a system of therapeutic diet menus, and it is used in various health and rehabilitation centers for people undergoing treatment or recovering from various diseases. The composition of the diet includes proteins—100 g (of which 65 g are animal ones), fats—100 g (of which 80 g are animal ones), carbohydrates—400 g (sugar—100 g), salt—12–15 g, free liquid—about 1.5 L. The total calorie intake is about 2500 kcal. The diet is enriched with vitamins. The most indigestible and spicy foods are excluded from the diet. The duration of this program is designed for 14 days in accordance with the order of the Ministry of Health of Russia dated 28 September 2020 N 1029n.

### 2.3. Sample Collection

Blood and fecal samples were collected from patients with post-COVID-19 syndrome on the day of admission (*n* = 39); repeatedly blood and fecal samples were collected from patients after rehabilitation for 14 days (*n* = 28) at the Federal Research and Clinical Center of Intensive Care Medicine and Rehabilitology (Moscow, Russia). The total number was 67 serum and 67 fecal samples from patients. The blood samples from the healthy volunteers (*n* = 48) were collected in Federal State Budgetary Institution “N.N. Burdenko Main Military Clinical Hospital” (Moscow, Russia). Blood samples were collected from a peripheral vein into EDTA tubes for the complete blood count, into 3.2% sodium citrate tubes for the coagulation test and into anticoagulant-free test tubes for the other purposes. Serum samples were obtained by blood centrifuging at 1500× *g* for 10 min on the same day. Serum aliquots were poured into disposable Eppendorf tubes, frozen, and stored at −35 °C. Fecal samples were collected into disposable sterile containers. The time from sample collection to fecal sample analysis did not exceed 12 h.

### 2.4. Sample Analysis

A complete blood count, including white blood cell count, red blood cell count, hemoglobin, hematocrit, and other parameters, was performed by a hematology analyzer (UniCel DxH800, Beckman Coulter, Brea, CA, USA). Coagulation test including D-dimer was performed on a coagulation analyzer (Sysmex CS-2000i, Kobe, Japan). A blood biochemistry test, including levels of bilirubin, total protein, creatinine, glucose, cholesterol, lactate dehydrogenase (LDH), alanine transaminase (ALT), aspartate transaminase (AST), C-reactive protein (CRP), and uric acid, was performed by a chemistry analyzer (AU480, Beckman Coulter, Brea, CA, USA). Biomarkers, including interleukin-6 (IL-6) and neuron-specific enolase (NSE), were analyzed using electrochemiluminescence (Cobas e411, Roche, Basel, Switzerland). Metabolites, including benzoic, phenylpropionic, phenyllactic, 4-hydroxybenzoic, 4-hydroxyphenylacetic, homovanillic, 4-hydroxyphenylpropionic, 4-hydroxyphenyllactic, succinic, and fumaric acids, were measured using gas chromatography-mass spectrometry (Trace GC 1310 gas chromatograph and ISQ LT mass spectrometer, Thermo Electron Corporation, Santa Clara, CA, USA) [14].

Fecal sample preparation and DNA extraction were previously described [15]. The composition of the gut microbiota was analyzed using Colonoflor-16 (biocenosis) kits (AlphaLab, St. Petersburg, Russia) by real-time PCR detection (CFX 96, BioRad, Hercules, CA, USA). The Operating Instructions for Colonoflor-16 (biocenosis) kit, obtained from AlphaLab, Russia, are demonstrated in Appendix A. Reference values (RV) were obtained for the healthy volunteers without gastrointestinal complaints (age over 14 years) from the kit instructions.

### 2.5. Statistical Analysis

The Shapiro–Wilk test was used to assess the normality of the data distribution, and it was revealed that all quantitative indicators of parametric comparison criteria were inapplicable due to the small number of outcomes, and the comparative intergroup analysis was carried out using nonparametric statistics. Independent group differences were explored using the Mann–Whitney U test. The Wilcoxon signed-rank test was used to compare two related groups. Correlation analysis was carried out using Spearman’s nonparametric correlation coefficient. The differences and correlations were considered significant at *p* < 0.05. All analyzes were done using IBM SPSS Statistics for Windows, Version 27.0. (Armonk, NY, USA: IBM Corp.). The data in the tables are described using median and interquartile ranges, minimum and maximum levels.

## 3. Results

### 3.1. Clinical Condition of Patients

From their medical history, it could be seen that all patients had evidence of lung involvement on CT ranging from 10 to 75% (stage I—40%, II—35%, III—15, IV—10%) during acute COVID-19, and that 23% of patients received antibacterial therapy. On the day of admission to rehabilitation, all patients (*n* = 39) were examined by the therapist using a questionnaire for COVID-19 survivors on admission (Appendix A). Oxygen saturation levels (SpO_2_) were measured using a pulse oximeter and ranged from 95 to 96%. All patients had the following post-COVID-19 syndrome symptoms: shortness of breath, asthenic syndrome, rapid fatigue during little physical exertion, general weakness, and fatigue; 35% of patients had sleep disorders, 10% of patients reported frequent headaches and cough, 5% of patients reported arrhythmia and myalgia. These symptoms appeared after suffering from COVID-19.

Of the 39 patients included in our study, only 28 were treated with the 14-day rehabilitation program. After the 14-day rehabilitation program, patients (*n* = 28) were repeatedly examined by the therapist. They noted a decrease in shortness of breath after physical activity and an improvement in state of health. The patients’ oxygen saturation levels (SpO_2_) increased and ranged from 97 to 99%. According to the results of the Wilcoxon signed-rank test, the increase in the saturation during 14 days of the rehabilitation was statistically significant (*p* < 0.01) but these changes could be within the pulse oximeter measurement error (2% error in the range of oxygen saturation level 90–100%). In most patients, vesicular breathing was carried out in all parts of the lungs. Some patients (*n* = 20) agreed to CT scan examination after the rehabilitation (Appendix A) and positive dynamics in lung injury (%) were observed comparing to acute COVID-19 (*p* < 0.001, the Wilcoxon signed-rank test).

### 3.2. Laboratory Parameters

A complete blood count (Appendix A), coagulation test (Appendix A), blood chemistry and biomarkers (Appendix A) were done for all patients on the day of admission (*n* = 39). The results on admission and after rehabilitation for the patients who were treated for 14 days (*n* = 28) are demonstrated in Table 1, Table 2 and Table 3.

The erythrocyte sedimentation rate was elevated more often than other parameters (in 10 of 39 patients, 25%). Red blood cell count, hemoglobin, and hematocrit were elevated in 6–9 of 39 patients, 15–23%. Other parameters of the complete blood count were out of the corresponding reference ranges in less than 10% of patients (Appendix A).

After 14 days of rehabilitation, there were no statistically significant changes in the results of the complete blood count, except for hemoglobin and hematocrit, although these parameters remained elevated in 21% of patients (Table 1).

All parameters of the coagulation test, including D-dimer, were out of reference ranges on admission in 5–8 of 39 patients, 13–20% (Appendix A). Only prothrombin time statistically changed after 14 days of rehabilitation and remained elevated in 2 of 28 patients (Table 2). Moreover, in some patients (*n* = 20) who agreed to CT scan examination after the rehabilitation (Appendix A), a positive correlation between lung injury (%) and D-dimer was observed—*r* = 0.67.

The values of the total protein, creatinine, lactate dehydrogenase, alanine transaminase, and aspartate transaminase were within the reference values for most patients (>90%), including parameters that were found to be statistically different after 14 days of rehabilitation (Table 3). Bilirubin and glucose were elevated in 5–7 of 39 patients, 13–18%; C-reactive protein was elevated in 8 of 39 patients, 20%; uric acid was elevated in 10 of 39 patients, 25%; cholesterol was elevated in 19 of 39 patients, 49% (Appendix A). The data about elevated glucose, uric acid, and cholesterol correlated with the information from the patients’ medical history and comorbidities (diabetes, gout, atherosclerosis). In some patients (*n* = 20) who agreed to CT scan examination after the rehabilitation (Appendix A), a positive correlation between lung injury (%) and glucose was observed—*r* = 0.51.

Two specific biomarkers were measured in serum samples using electrochemiluminescence. NSE levels were within reference values for most patients, although they were found to be statistically different at the two time points. IL-6 levels were higher than the reference value in 85% of patients at both time points (Table 3 and Appendix A).

### 3.3. Low-Molecular-Weight Metabolites in Serum

On the day of admission for the rehabilitation, the serum profile of some low-molecular-weight metabolites of all patients (*n* = 39) was compared with that of healthy volunteers (Appendix A). The following metabolites were measured using gas chromatography-mass spectrometry: metabolites of tyrosine, such as 4-hydroxybenzoic, 4-hydroxyphenylacetic, homovanillic, 4-hydroxyphenylpropionic, and 4-hydroxyphenyllactic acids; metabolites of phenylalanine, such as phenylpropionic, phenyllactic, and benzoic acids; metabolites of Krebs cycle, such as succinic and fumaric acids. Table 4 and Appendix A demonstrate the metabolites that were quantitatively measured in serum samples (concentration was equal or more than 0.5 µmol/L with relative standard deviation less than 30%). The concentrations of 4-hydroxybenzoic, succinic, and fumaric acids were statistically different in patients compared to healthy volunteers (*p* < 0.001). Moreover, 4-Hydroxybenzoic acid was not quantified in any sample from healthy volunteers, but it was measured in all samples of patients with the medial value of more than 6 µmol/L. Succinic and fumaric acids were detected in all serum samples of patients and healthy volunteers, however, their median values were 2.5 and 2 times higher in patients, respectively (Appendix A).

The profile of the low-molecular-weight metabolites was revealed in patients (*n* = 28) after 14 days of the rehabilitation (Table 4). The level of only one metabolite 4-hydroxybenzoic acid statistically reduced and the median value decreased by 2 times, but it was still higher than in healthy volunteers with median value of 3 µmol/L. The levels of succinic and fumaric acids did not change and remained higher than in healthy volunteers. The level of the phenylpropionic acid increased in dynamics and was quantitatively measured in 15 of 28 patients after rehabilitation in comparison with 8 of 28 patients on admission.

### 3.4. Gut Microbiota Taxonomy

Taxonomic abundance of the gut microbiota in patients with post-COVID-19 syndrome (*n* = 39) was evaluated using real-time PCR (Appendix A). Most patients were characterized by high levels of total bacterial mass (61%) and *Bacteroides* spp. (95%), and 36% of the patients had a high value of *Bacteroides fragilis*/*Faecalibacterium prausnitzii* ratio. *Bifidobacterium* spp. (69%), *Enterobacter* spp. (67%), *Escherichia coli* (56%), *Staphylococcus aureus* (28%), *Fusobacterium nucleatum* (18%); *Parvimonas micra*, *Citrobacter* spp., *Proteus vulgaris/mirabilis*, *Clostridium perfringens*, and *Escherichia coli enteropathogenic* were elevated in less than 13% of patients.

Nine of 39 patients were treated with antibiotics during the acute phase of COVID-19 based on medical records. Antibiotics were cephalosporins—44%, broad spectrum beta-lactamase penicillins—33%, fluoroquinolones—22%, macrolides—11%, and carbapenems—11%; in two patients, broad spectrum beta-lactamase penicillins and cephalosporins, or carbapenems and fluoroquinolones, were used. On admission, these 9 patients had a high level of total bacterial mass, in 2 patients *Lactobacillus* spp. were decreased; in 4 patients an increased level of *Bacteroides fragilis*/*Faecalibacterium prausnitzii* ratio (>100) was observed, and only in 3 patients *Akkermansia muciniphila* was detected.

*Bifidobacterium* spp. and *Escherichia coli* were subsequently decreased after rehabilitation (*p* = 0.019 and 0.023, respectively) (Table 5). *Enterococcus* spp., *Escherichia coli* enteropathogenic, *Citrobacter* spp., and *Fusobacterium nucleatum* had positive dynamics after 14 days. One of the two patients who had *Escherichia coli enteropathogenic* and the patient who had *Enterococcus* spp. on admission had a decrease in IL-6, 4-hydroxybenzoic and *Bacteroides fragilis*/*Faecalibacterium prausnitzii* ratio (<100) after 14 days of rehabilitation. In another patient, in whom *Escherichia coli enteropathogenic* was detected on admission, there was no positive dynamics in these indicators after rehabilitation.

### 3.5. COVID-19 Vaccination

Twenty of 39 patients were vaccinated prior to being infected with COVID-19. We divided all patients into two groups and revealed the statistically significant differences between them using all analyzed parameters from Section 3.2, Section 3.3 and Section 3.4. Four parameters appeared to be statistically different between these two groups on admission (Table 6). The glucose and *Bacteroides fragilis*/*Faecalibacterium prausnitzii* ratio were more often within the reference values in vaccinated patients.

## 4. Discussion

The main results of our study describe a number of different variables that were found to be out of the reference ranges in the post-COVID-19 patients. The study involved patients (*n* = 39) who were admitted to the rehabilitation center with a number of post-COVID-19 syndrome symptoms including shortness of breath, fatigue, sleep disturbances, etc. Some patients (*n* = 28) were enrolled in a 14-day post-COVID-19 rehabilitation program mostly aimed at restoration of the respiratory system using breathing exercises. In general, all patients (*n* = 28) noted an improvement in overall health and respiratory function after 14 days of rehabilitation. Despite that the results of the complete blood count, coagulation, and blood chemistry tests were within the reference ranges for most patients (Table 1, Table 2 and Table 3), the elevated levels of the proinflammatory IL-6 (Table 3), 4-hydroxybenzoic, succinic, and fumaric acids (Table 4) were found in most patients. Moreover, gut dysbiosis was observed in patients’ feces (Table 5). These results were obtained on the day of admission and did not reach the reference values after 14 days of rehabilitation. The changes in hemoglobin, hematocrit, prothrombin time, lactate dehydrogenase, alanine transaminase, neuron-specific enolase, 4-hydroxybenzoic acid, Bifidobacterium spp., and Escherichia coli were statistically significant after the rehabilitation program (*p* < 0.05). However, the absence of the correlation of these variables with the lung injury (Appendix A) in some patients (*n* = 20) who agreed to CT scan examination, indicates that these variables were not associated with the improvement in respiratory function of the patients after rehabilitation.

A systematic review and meta-analysis of 24 inflammatory and vascular biomarkers in post-COVID-19 syndrome [6] provided the information about high levels of C-reactive protein, D-dimer, lactate dehydrogenase, leukocytes, and lymphocytes in patients with post-COVID-19 syndrome compared to those without it. C-reactive protein was pointed out as a potential diagnostic biomarker for post-COVID-19 syndrome in another systematic review [7]. C-reactive protein and lactate dehydrogenase were higher in the post-COVID-19 patients independent from the severity of the disease and lung fibrotic areas [3]. Higher D-dimer, C-reactive protein, and erythrocyte sedimentation rate and lower hemoglobin were detected in post-COVID-19 patients in another study [2]. In our study, D-dimer, lactate dehydrogenase, leukocytes, and lymphocytes were within their reference ranges in most patients (>85%) and did not demonstrate clinical significance. However, a positive correlation between D-dimer and lung injury was found. Erythrocyte sedimentation rate and C-reactive protein were elevated in 25 and 20% of patients, respectively, and could be considered as useful biomarkers for the treatment of post-COVID-19 patients.

IL-6 is a pro-inflammatory cytokine that is associated with the acute phase of inflammation. It increases in cases of infection, inflammation, or trauma [16] and can affect the neurons and intensify neuroinflammation [17]. High IL-6 levels were previously described in different original studies on post-COVID-19 patients [4,5] and summarized in different reviews [6,7,8]. According to a recent study, ambulatory elderly patients with heart failure and elevated IL-6 more frequently presented with atrial fibrillation, hypercholesterolemia, diabetes mellitus, anemia, and renal dysfunction. Moreover, higher mortality was observed in stable heart failure patients with elevated IL-6 [18]. Elevated IL-6 levels with median value of 12 pg/mL were observed in our study in 85% of patients. Most of our patients are elderly. Thus, having elevated IL-6, our patients can be at a risk group of different comorbidities. Moreover, IL-6 is considered as a new therapeutic target as it plays a pleiotropic role in activating the inflammatory response [19] and the use of the tocilizumab, a humanized anti-IL-6 receptor antibody, was associated with a lower risk of mortality and mechanical ventilation requirement among COVID-19 patients [20]. We can conclude that the monitoring of the IL-6 level could be a therapeutic target in the treatment of the cause of inflammation in post-COVID-19 patients.

NSE is one of the indicators of brain cell damage. Its high levels were reported in acute COVID-19 patients and summarized in a review [21]. Despite that high NSE levels were associated with axonal or lung injury and neuroinflammation, its role in COVID-19 infection and in post-COVID-19 syndrome is unclear. In our study, the NSE levels were within the reference range for most patients (>90%) and did not demonstrate clinical significance despite the statistically significant difference in post-COVID-19 patients at two time points.

Plasma metabolic profiling in post-COVID-19 patients was described in several studies. One of the studies revealed elevated taurine and reduced glutamine/glutamate ratio in both the acute-phase and in three-month post-acute COVID-19 syndrome [10]. Another study reported alterations in 16 metabolites and 74 lipoprotein compounds in acute and post-COVID-19 patients. The metabolites included amino acids (phenylalanine, tyrosine, valine, methionine), ketone bodies (3-hydroxybutyrric and acetoacetic acids, acetone), isoleucine, mannose, lactic and acetic acids [9]. We provided the metabolic profiling of the serum samples of the post-COVID-19 patients that was focused on the metabolites of the phenylalanine and tyrosine and mitochondrial metabolites. The high levels of the metabolites of the Krebs cycle succinic and fumaric acids were observed in the post-COVID-19 patients in our study. High levels of mitochondrial metabolites in the blood are attributed to a shift of the general metabolism. High levels of the succinic acid could be explained by tissue hypoxia or inflammation [22]. One of the figures in the mentioned above metabolic profiling study [9] contained the information about succinic acid without any comments in the text. The figure demonstrated statistically insignificant lower concentration of succinic acid in serum samples from acute compared to post COVID-19 patients. Perhaps, this result could be statistically significant in case of the comparison with healthy people. We suppose that the decrease of the mitochondrial metabolites to the normal values could indicate an improvement of the metabolism in post-COVID-19 patients.

We hypothesized that some microbial metabolites, which were previously elevated in different groups of patients with infectious complications in the intensive care units, could be elevated in comparison with healthy volunteers [23,24]. These metabolites, i.e., phenyllactic, 4-hydroxyphenylacetic, and 4-hydroxyphenyllactic acids, were not statistically elevated in the post-COVID-19 patients. These results indicate that the patients included in the study did not have a clinically significant bacterial infection.

The level of the phenylpropionic acid, one of the phenylalanine metabolites, was quantitatively measured in 8 of 28 patients on admission. After rehabilitation, it increased in 7 of 28 patients and finally was measured in 15 of 28 patients, 53%. This metabolite is associated with “healthy” microbiota; usually detected in healthy people and reduced in different groups of patients [25]. Thus, an increase in phenylpropionic acid in 7 patients could indicate the process of restoring the “healthy” composition of the microbiota.

The level of 4-hydroxybenzoic acid was elevated in all patients, which we had not previously detected in either patients or healthy volunteers. Moreover, 4-Hydroxybenzoic acid is produced in bacteria, plants, and humans. In humans, 4-hydroxybenzoic acid could come from plant-based diets or could be produced through microbial fermentation of aromatic amino acids in the colon [26]. Further, it is involved in ubiquinone biosynthesis [27] or conjugated into sulfate and glucuronide conjugates [28] and 4-Hydroxybenzoic acid was reported as statistically high (11–16 fold) in patients with skin cancers in comparison with healthy volunteers, but no concentrations were provided [29]. According to the data from healthy volunteers and our previous studies, we suppose that the concentration of this acid is normally lower than 0.5 µmol/L (Table 4).

Moreover, 4-Hydroxybenzoic acid can be biosynthesized by chorismate lyase, which catalyzes the first step in ubiquinone biosynthesis in *E. coli* and other gram-negative bacteria [30]. In our study, the concentration of *E. coli* in feces was higher than the reference value in 17 of 28 patients (60%) involved in the rehabilitation program (Table 5). It statistically decreased after the 14-day rehabilitation (*p* = 0.023) and remained high in 6 of 28 patients (21%). The statistically significant decrease of 4-hydroxybenzoic acid in the serum was also observed in patients involved in the 14-day rehabilitation program (*p* = 0.003). Despite that no statistically significant correlation was found between the dynamics of concentrations of the 4-hydroxybenzoic acid and *E. coli*, the decrease in both fecal *E. coli* and serum 4-hydroxybenzoic acid was observed in 13 of 28 patients (46%). This result indicates a potentially beneficial effect of rehabilitation programs on the gut microbiota in patients with post-COVID-19 syndrome. Moreover, we attribute the absence of the high values of 4-hydroxybenzoic acid in our previous studies in different groups of patients to the antimicrobial therapy that is usually prescribed to most surgical or critically ill patients and that affects the concentration of *E. coli.*

Various laboratory markers of dysbiosis, such as an excess of bacterial mass, a high ratio of the *Bacteroides fragilis*/*Faecalibacterium prausnitzii* group (a sign of anaerobic imbalance), an excess of the number of pro-inflammatory taxa were detected in almost all patients. We observed similar changes in patients with a prolonged stay in intensive care with long courses of antimicrobial therapy [31]. *Faecalibacterium prausnitzii* with known immunomodulatory potential was underrepresented in COVID-19 patients and remained low in samples collected up to 30 days after COVID-19 resolution [32]. In 16 of 28 patients (%) in our study, *Faecalibacterium prausnitzii* was higher than the reference value at both time points, indicating the restoration of the gut microbiota.

Some specific *Bacteroides* species capable of downregulating ACE2 expression in the mouse gut are inversely correlated with SARS-CoV-2 burden [33]. Metformin-treated patients with type 2 diabetes and COVID-19 without antibiotic treatment showed increased *Bacteroides* spp. compared to those with antibiotic treatment [34]. Based on these results, and the anti-inflammatory properties of *Bacteroides*, it can be hypothesized that their increase plays a role in controlling inflammation through the gut–lung axis.

Antibiotic treatment may shift the gut microbiome composition towards opportunistic bacteria, particularly *Enterococcus*. COVID-19 patients with increased IL-6, D-dimer, and ferritin levels receiving antibiotic treatment were more likely to show dysbiosis with increased abundance of *Enterococcus* [35]. One of the markers of gut dysbiosis can be *F. nucleatum*, which is generally considered to be in the oral cavity [36]. *F. nucleatum* has been shown to colonize colon mucus with associated mucosal inflammation [37]. *F. nucleatum* bacteremia has been shown to occur in 4 patients with SARS-CoV-2, apparently due to a translocation due to the inflammatory response [38]. In our study, *Enterococcus* spp. and *F. nucleatum* levels were elevated in 1 and 6 patients, respectively, on admission; they had positive dynamics after 14 days of rehabilitation (*F. nucleatum* was detected in 3 patients, *Enterococcus* spp. was not detected in any case).

Remarkably, gastrointestinal dysbiosis after COVID-19 can occur, even in the absence of gastrointestinal symptoms [33]. This indicates the need for laboratory tests to identify the features of dysbiosis after the disease with subsequent correction. The impact of diet, nutrients, and probiotics in reducing the severity of COVID-19 infection has been suggested [11,39,40]. One of the promising measures aimed at correcting the microbiota is a complex phage therapy, which has proven itself in a pilot study with the decrease of *Bacteroides fragilis*/*Faecalibacterium prausnitzii* ratio of anaerobic dysbiosis and no side effects in post-COVID-19 rehabilitation and in chronically critically ill patients [15,41]. A study managing COVID-19 with complex oral probiotics in addition to standard treatment showed remission of gastrointestinal symptoms for nearly all patients compared to less than half of the control [42].

Our study had certain limitations. The first is a small cohort of patients and further studies with larger cohorts should be conducted to confirm our findings. Because of the small cohort of patients, the influence of the patients’ comorbidities, the acute-COVID-19 treatment and other factors were not evaluated. There is also no control group of patients with post-COVID-19 syndrome who did not receive any rehabilitation procedures during the same time period to show that our findings are not related only to time.

Another limitation is the insufficient screening of patients on admission to rehabilitation. The use of the Gastrointestinal Symptom Rating Scale (GSRS) and the Bristol Stool Scale may be helpful in assessing clinical signs of possible microbiota dysfunction. The change in gut-related variables in our study could just be due to the change in patients’ diet during rehabilitation and were not necessarily related to the COVID-19 disease or related to other disease outcomes. In addition, patients were not assessed for cognitive function and neurological dysfunction and were not examined on the day of discharge with the same questionnaire (Appendix A) as on admission.

Despite the statistically significant differences in some variables in vaccinated and unvaccinated patients (Table 6), we cannot identify the role of vaccination in such differences, since our patients may have had them before vaccination. However, there are studies describing the association of vaccination with post-COVID-19 symptoms [43,44], and this may be the subject for further research.

Patients with extra-pulmonary manifestations during acute COVID-19 might signify a more inflammatory response, especially when more than one system is involved [45], and this may affect our results, but such additional information was not obtained from patients and we were unable to evaluate its effect on our results.

Another limitation is the use of real-time PCR to characterize the gut microbiota instead of 16s rRNA sequencing or whole genome sequencing, which limits the taxa that can be detected and does not allow for the evaluation of minor bacterial species. At the same time, real-time PCR provides the ability to track changes and identify major nosocomial pathogens virtually at the patient’s bedside in routine clinical practice.

## 5. Conclusions

Rehabilitation of patients who have suffered from COVID-19 infection but do not feel completely healthy is an actual and unsolved issue nowadays. It is not enough to explain post-COVID-19 syndrome by the exacerbation of chronic diseases. In this study, indeed, alterations in some biochemical and metabolic parameters were detected in a number of patients, which can be explained by the presence of concomitant diseases. At the same time, in patients with post-COVID-19 syndrome, along with an increase in inflammatory markers, alterations in the composition of the gut microbiota were detected. It turned out that gut dysbiosis persisted and did not improve during the 14-day standard rehabilitation program, as also interleukin-6, mitochondrial metabolites, and some other markers, which continued to be out of reference ranges. We suppose that clinical symptoms of post-COVID-19 syndrome can be considered as manifestations of the microbiota dysfunction at the level of the whole human body, and the post-COVID-19 rehabilitation program should be personalized, taking into account not only the baseline levels of standard laboratory parameters, but also the taxonomy and function of the gut microbiota. Further research should be aimed at exploring possible methods of influencing the symptoms of post-COVID-19 syndrome, including special exercises, specific diets, pre-, pro-, and metabiotic, etc.

## Figures and Tables

**Table 1 jpm-13-00971-t001:** The results of the complete blood count in patients with post-COVID-19 syndrome (*n* = 28) on the day of admission and after 14 days of rehabilitation, and the results of the Wilcoxon signed-rank test. The statistically significant differences are highlighted in bold. Reference values are combined for male/female, but *n* (c > RV)/*n* (c < RV) indicates the number of samples with a higher/lower level than the corresponding reference value taking into account the difference in the reference values for male/female.

Parameter	Reference Values	Patients on Admission (*n* = 28)	Patients after 14 Days (*n* = 28)	*p*-Value
White Blood Cell Count (WBC), ×10^9^/L	3.8–11.8	5.6 (5.3–6.8), 3.0–10.9 *n* (c < RV) = 1	5.6 (4.9–6.1), 3.2–12.0 *n* (c > RV) = 1 *n* (c < RV) = 1	0.946
Red Blood Cell Count (RBC), ×10^12^/L	3.63–5.63	4.56 (4.18–5.15), 3.71–5.56 *n* (c > RV) = 6	4.74 (4.33–4.99), 3.99–5.50 *n* (c > RV) = 4	0.060
Hemoglobin (Hb), g/L	109–163	134 (125–150), 95–169 *n* (c > RV) = 7 *n* (c < RV) = 1	140 (129–147), 95–163 *n* (c > RV) = 6 *n* (c < RV) = 1	**0.027**
Hematocrit (Hct), %	31.2–47.1	40.6 (37.2–44.7), 31.1–49.6 *n* (c > RV) = 7 *n* (c < RV) = 1	42.6 (39.1–43.6), 30.7–48.1 *n* (c > RV) = 6 *n* (c < RV) = 1	**0.027**
Mean Cell Volume (MCV), fL	75.5–95.3	89.4 (86.2–92.4), 62.2–96.9 *n* (c > RV) = 3 *n* (c < RV) = 1	88.2 (86.5–92.6), 62.8–97.9 *n* (c > RV) = 2 *n* (c < RV) = 1	0.840
Mean Cell Hemoglobin (MCH), pg/cell	24.7–33.4	29.8 (28.5–30.9), 19.0–33.7 *n* (c > RV) = 1 *n* (c < RV) = 1	29.5 (28.8–30.7), 19.4–33.5 *n* (c > RV) = 1 *n* (c < RV) = 1	0.961
Mean Cell Hemoglobin Concentration (MCHC), g/L	323–356	333 (326–336), 305–348 *n* (c < RV) = 3	332 (330–336), 310–342 *n* (c < RV) = 3	0.807
Red Blood Cell Distribution Width (RDW), %	12.3–17.7	14.1 (13.0–15.0), 10.2–20.7 *n* (c > RV) = 1 *n* (c < RV) =2	14.25 (13.2–14.7), 12.4–21.2 *n* (c > RV) = 1	0.893
Platelet Count (Plt), ×10^9^/L	179–408	232 (201–295), 72–436 *n* (c > RV) = 1 *n* (c < RV) = 3	249 (206–287), 140–437 *n* (c > RV) = 1 *n* (c < RV) = 3	0.162
Mean Platelet Volume (MPV), fL	7.9–10.8	9.3 (8.4–9.9), 7.5–10.8 *n* (c < RV) = 1	8.9 (8.4–9.7), 7.3–11.0 *n* (c < RV) = 1	0.807
Neutrophil, %	42.7–76.8	55.2 (48.1–59.5), 37.4–71.3 *n* (c < RV) = 2	54.3 (47.6–57.3), 34.9–74.6 *n* (c < RV) = 3	0.809
Lymphocytes, %	16.0–45.9	32.4 (29.1–40.2), 18.6–48.8 *n* (c > RV) = 3	33.4 (30.9–41.4), 12.1–52.5 *n* (c > RV) = 2 *n* (c < RV) = 1	0.809
Monocytes, %	4.3–10.9	8.3 (6.9–9.4), 5.9–12.7 *n* (c > RV) = 3	8.0 (6.9–10.1), 5.2–16.3 *n* (c > RV) = 2	0.764
Eosinophils, %	0.5–7.0	2.5 (1.5–3.6), 1.1–8.8 *n* (c > RV) = 1	2.3 (1.8–3.6), 0.9–13.3 *n* (c > RV) = 1	0.627
Basophil, %	0.2–1.3	0.7 (0.6–0.9), 0.1–1.7 *n* (c > RV) = 2 *n* (c< RV) = 1	0.9 (0.6–1.1), 0.4–5.3 *n* (c > RV) = 2	0.321
Absolute Neutrophil, ×10^9^/L	1.9–8.2	3.3 (2.6–3.9), 1.7–6.2 *n* (c < RV) = 2	3.1 (2.4–3.6), 1.5–7.0 *n* (c < RV) = 2	0.493
Absolute Lymphocytes, ×10^9^/L	1.1–3.1	1.9 (1.6–2.5), 0.8–3.5 *n* (c > RV) = 2 *n* (c < RV) = 2	1.9 (1.6–2.4), 0.7–3.7 *n* (c > RV) = 2 *n* (c < RV) = 3	0.605
Absolute Monocytes, ×10^9^/L	0.2–0.9	0.5 (0.4–0.6), 0.3–0.8	0.5 (0.4–0.5), 0.3–1.0 *n* (c > RV) = 1	0.388
Absolute Eosinophils, ×10^9^/L	<0.5	0.2 (0.1–0.2), <0.1–0.6 *n* (c > RV) = 1	0.1 (0.1–0.2), <0.1–0.7 *n* (c > RV) = 1	0.110
Absolute Basophil, ×10^9^/L	<0.10	0.01 (<0.10–0.10), <0.10–0.10	<0.10 (<0.10–0.10), <0.10–0.30 *n* (c > RV) = 2	0.110
Erythrocyte Sedimentation Rate (ESR), mm/hr	<20	14 (8–23), 3–37 *n* (c > RV) = 7	12 (5–18), 2–40 *n* (c > RV) = 6	0.100

**Table 2 jpm-13-00971-t002:** The results of the coagulation test in patients with post-COVID-19 syndrome (*n* = 28) on the day of admission and after 14 days of rehabilitation, and the results of the Wilcoxon signed-rank test. The statistically significant differences are highlighted in bold. *n* (c > RV)/*n* (c < RV) indicates the number of samples with a higher/lower level than the corresponding reference value.

Parameter	Reference Values	Patients on Admission (*n* = 28)	Patients after 14 Days (*n* = 28)	*p*-Value
Prothrombin time (PT), sec	9.4–12.5	10.9 (10.4–11.8), 9.4–16.4 *n* (c > RV) = 4	11.0 (10.5–11.6), 9.2–15.5 *n* (c > RV) = 2 *n* (c < RV) = 1	**0.039**
Prothrombin by Quik %	80–140	118 (86–134), 66–167 *n* (c > RV) = 3 *n* (c < RV) = 2	115 (96–129), 75–173 *n* (c > RV) = 4 *n* (c < RV) = 2	0.224
International Normalised Ratio (INR)	0.90–1.20	1.03 (0.94–1.08), 0.86–1.50 *n* (c > RV) = 4 *n* (c < RV) = 1	1.01 (0.94–1.06), 0.85–1.42 *n* (c > RV) = 2 *n* (c < RV) = 1	0.073
Fibrinogen Activity, g/L	2.38–4.98	3.21 (2.55–3.44), 2.04–3.81 *n* (c < RV) = 3	3.10 (2.55–3.53), 2.22–4.61 *n* (c < RV) = 5	0.786
Activated Partial Thromboplastin Time (PTT), sec	25.0–36.5	29.0 (27.7–33.4), 23.3–41.4 *n* (c > RV) = 4 *n* (c < RV) = 2	28.7 (27.9–32.1), 23.8–39.4 *n* (c > RV) = 2 *n* (c < RV) = 1	0.306
Thrombin time (TT), sec	11.0–17.8	16.2 (13.4–17.6), 12.1–19.2 *n* (c > RV) = 5	16.9 (14.3–17.4), 13.1–18.7 *n* (c > RV) = 5	0.118
D-dimer, μ/mL	<0.49	0.21 (0.14–0.32), 0.07–1.89 *n* (c > RV) = 3	0.24 (0.19–0.41), 0.10–1.35 *n* (c > RV) = 4	0.085

**Table 3 jpm-13-00971-t003:** The results of the blood chemistry test, the levels of interleukin-6 and neuron-specific enolase in the serum samples of patients with post-COVID-19 syndrome (*n* = 28) on the day of admission and after 14 days of rehabilitation, and the results of the Wilcoxon signed-rank test. The statistically significant differences are highlighted in bold. *n* (c > RV) indicates the number of samples with a level above the corresponding reference value.

Parameter	Reference Values	Patients on Admission (*n* = 28)	Patients after 14 Days (*n* = 28)	*p*-Value
Bilirubin, μmol/L	5.0–21.0	14.3 (9.2–18.4), 5.4–38.1 *n* (c > RV) = 4	12.1 (9.0–14.5), 3.0–39.0 *n* (c > RV) = 3 *n* (c < RV) = 1	0.106
Total Protein, g/L	66.0–83.0	68.1 (66.8–72.4), 62.1–81.2 *n* (c < RV) = 1	71.3 (68.4–72.7), 61.3–75.6 *n* (c < RV) = 2	0.158
Creatinine, μmol/L	58.0–110.0	84.1 (79.6–92.4), 69.3–110.7 *n* (c > RV) = 1	87.2 (81.3–93.4), 62.8–120.4 *n* (c > RV) = 1	0.750
Glucose, mmol/L	4.1–5.9	5.3 (5.0–5.9), 4.1–9.1 *n* (c > RV) = 5	5.5 (5.0–6.0), 4.0–8.5 *n* (c > RV) = 7 *n* (c < RV) = 1	0.733
Cholesterol, mmol/L	< 5.2	5.4 (4.4–6.5), 2.4–8.4 *n* (c > RV) = 16	5.3 (4.3–6.3), 2.3–7.6 *n* (c > RV) = 14	0.234
Lactate Dehydrogenase (LDH), U/L	<247.0	194.0 (164.4–213.1), 131.4–306.3 *n* (c > RV) = 2	197.5 (166.3–226.8), 128.7–334.7 *n* (c > RV) = 3	**0.046**
Alanine Transaminase (ALT), U/L	<50.0	19.7 (14.1–28.4), 9.2–44.5 *n* (c > RV) = 2	19.2 (14.4–28.8), 8.8–68.0 *n* (c > RV) = 3	0.232
Aspartate Transaminase (AST), U/L	<50.0	20.3 (18.8–23.9),15.5–91.7 *n* (c > RV) = 3	22.6 (19.1–29.5), 14.0–121.4 *n* (c > RV) = 3	**0.005**
C-Reactive Protein (CRP), mg/L	<5.0	0.6 (0.1–0.9), 0.1–14.0 *n* (c > RV) = 4	0.6 (0.1–0.8), 0.1–8.4 *n* (c > RV) = 2	0.925
Uric acid, μmol/L	154.7–428.0	321.6 (256.0–386.7), 196.5–516.0 *n* (c > RV) = 8	321.1 (243.1–385.9),157.3–567.7 *n* (c > RV) = 6	0.524
Interleukin-6 (IL-6), pg/mL	<7.0	11.7 (8.1–15.5), 1.5–61.9 *n* (c > RV) = 23	12.2 (8.7–17.7), 2.7–58.2 *n* (c > RV) = 25	0.255
Neuron-specific Enolase (NSE), ng/mL	<16.0	9.3 (4.9–11.8), 0.1–22.0 *n* (c > RV) = 2	9.6 (5.7–13.0), 0.1–48.0 *n* (c > RV) = 4	**0.038**

**Table 4 jpm-13-00971-t004:** The concentrations of metabolites in the serum samples of the healthy volunteers (*n* = 48) and patients with post-COVID-19 syndrome (*n* = 28) on the day of admission and after 14 days of rehabilitation, and the results of the Wilcoxon signed-rank test. The statistically significant differences are highlighted in bold. *n* (c > 0.5) indicates the number of samples with a level above the limit of quantitation.

Acid, µmol/L	Healthy Volunteers (*n* = 48)	Patients on Admission (*n* = 28)	Patients after 14 Days (*n* = 28)	*p*-Value
Benzoic	<0.5 (<0.5–<0.5), <0.5–0.6 *n* (c > 0.5) = 2	<0.5 (<0.5–<0.5), <0.5–0.8 *n* (c > 0.5) = 6	<0.5 (<0.5–0.5), <0.5–1.0 *n* (c > 0.5) = 7	-
Phenylpropionic	<0.5 (<0.5–0.5), <0.5–3.0 *n* (c > 0.5) = 15	<0.5 (<0.5–0.5), <0.5–2.4 *n* (c > 0.5) = 8	0.5 (<0.5–0.8), <0.5–2.0 *n* (c > 0.5) = 15	-
Phenyllactic	<0.5 (<0.5–<0.5), <0.5–0.7 *n* (c > 0.5) = 2	not detected	<0.5 (<0.5–<0.5), <0.5–0.5 *n* (c > 0.5) = 1	-
4-Hydroxybenzoic	not detected	6.8 (5.2–8.6), 2.0–13.0 *n* (c > 0.5) = 28	3.3 (1.5–6.7), 1.1–13.6 *n* (c > 0.5) = 28	**0.003**
4-Hydroxyphenylacetic	<0.5 (<0.5–<0.5), <0.5–1.2 *n* (c > 0.5) = 5	<0.5 (<0.5–0.5), <0.5–1.6 *n* (c > 0.5) = 8	<0.5 (<0.5–<0.5), <0.5–0.8 *n* (c > 0.5) = 6	-
4-Hydroxyphenyllactic	1.2 (0.9–1.5), 0.7–2.5 *n* (c > 0.5) = 48	1.1 (0.9–1.3), 0.7–2.7 *n* (c > 0.5) = 28	1.1 (0.9–1.4), 0.6–2.5 *n* (c > 0.5) = 28	0.908
Succinic	4.8 (4.4–6.0), 3.3–12.4 *n* (c > 0.5) = 48	14.0 (10.0–15.5), 8.0–25.0 *n* (c > 0.5) = 28	12.0 (9.2–16.0), 6.0–29.0 *n* (c > 0.5) = 28	0.181
Fumaric	1.3 (1.1–1.5), 0.8–2.3 *n* (c > 0.5) = 48	2.4 (1.9–3.0),1.5–5.5 *n* (c > 0.5) = 28	1.8 (1.6–2.2),1.3–11.6 *n* (c > 0.5) = 28	0.121

**Table 5 jpm-13-00971-t005:** Taxonomic abundance of the gut microbiota in patients with post-COVID-19 syndrome (*n* = 28) on the day of admission and after 14 days of rehabilitation, and the results of the Wilcoxon signed-rank test. The statistically significant differences are highlighted in bold. *n* (c > 10^4^/10^5^) indicates the number of samples with a level above the limit of quantitation. Reference values (RV) were obtained for the healthy volunteers without gastrointestinal complaints (age over 14 years). *n* (c > RV)/*n* (c < RV) indicates the number of samples with a higher/lower level than the corresponding reference value.

Parameter, lg CFU/g	Reference Values	Patients on Admission (*n* = 28)	Patients after 14 Days (*n* = 28)	*p*-Value
Total bacterial mass	10^11^–10^13^	2 × 10^13^ (1 × 10^13^–5 × 10^13^), 6 × 10^11^–4 × 10^15^ *n* (c > 10^4^) = 28 *n* (c > RV) = 17	2 × 10^13^ (1 × 10^13^–3 × 10^13^), 2 × 10^12^–1 × 10^14^ *n* (c > 10^4^) = 28 *n* (c > RV) =16	0.327
*Lactobacillus* spp.	10^7^–10^8^	4 × 10^7^ (3 × 10^6^–5 × 10^8^), 1 × 10^5^–4 × 10^10^ *n* (c > 10^5^) = 28 *n* (c > RV) = 11 *n* (c < RV) = 10	2 × 10^7^ (5 × 10^6^–1 × 10^8^), 1 × 10^5^–4 × 10^9^ *n* (c > 10^5^) = 28 *n* (c > RV) = 7 *n* (c < RV) = 10	0.311
*Bifidobacterium* spp.	10^9^–10^10^	3 × 10^10^ (9 × 10^9^–2 × 10^11^), 2 × 10^8^–3 × 10^12^ *n* (c > 10^5^) = 28 *n* (c > RV) = 19 *n* (c < RV) = 1	2 × 10^10^ (3 × 10^9^–7 × 10^10^), 8 × 10^7^–3 × 10^11^ *n* (c > 10^5^) = 28 *n* (c > RV) = 16 *n* (c < RV) = 4	**0.019**
*Escherichia coli*	10^6^–10^8^	3 × 10^8^ (3 × 10^7^–2 × 10^9^), 3 × 10^6^–5 × 10^11^ *n* (c > 10^5^) = 28 *n* (c > RV) = 17	6 × 10^7^ (3 × 10^7^–1 × 10^8^), 9 × 10^5^–1 × 10^10^ *n* (c > 10^5^) = 28 *n* (c > RV) = 6 *n* (c < RV) = 1	**0.023**
*Bacteroides* spp.	10^9^–10^12^	2 × 10^13^ (1 × 10^13^–5 × 10^13^), 6 × 10^11^–4 × 10^15^ *n* (c > 10^4^) = 28 *n* (c > RV) = 27	2 × 10^13^ (1 × 10^13^–3 × 10^13^), 2 × 10^12^–1 × 10^14^ *n* (c > 10^4^) = 28 *n* (c > RV) = 28	0.327
*Faecalibacterium prausnitzii*	10^8^–10^11^	4 × 10^11^ (6 × 10^10^–6 × 10^11^), 1 × 10^7^–5 × 10^13^ *n* (c > 10^4^) = 28 *n* (c > RV) = 16 *n* (c < RV) = 1	2 × 10^11^ (6 × 10^10^–6 × 10^11^), 1 × 10^10^–3 × 10^13^ *n* (c > 10^4^) = 28 *n* (c > RV) = 15	0.524
*Bacteroides thetaiotaomicron*	Any quantity is allowed	9 × 10^8^ (<10^5^–9 × 10^9^), <10^5^–1 × 10^11^ *n* (c > 10^5^) = 20	8 × 10^8^ (7 × 10^6^–4 × 10^9^), <10^5^–6 × 10^10^ *n* (c > 10^5^) = 22	0.833
*Akkermansia muciniphila*	<10^11^	<10^5^ (<10^5^–<10^5^), <10^5^–2 × 10^7^ *n* (c > 10^5^) = 4	<10^5^ (<10^5^–5 × 10^4^), <10^5^–6 × 10^8^ *n* (c > 10^5^) = 7	-
*Enterococcus* spp.	<10^8^	<10^5^ (<10^5^–<10^5^), <10^5^–4 × 10^12^ *n* (c > 10^5^) = 1 *n* (c > RV) = 1	not detected	-
*Escherichia coli enteropathogenic*	<10^4^	<10^4^ (<10^4^–<10^4^), <10^4^–5 × 10^6^ *n* (c > 10^4^) = 2 *n* (c > RV) = 2	not detected	-
*Candida* spp.	<10^4^	<10^4^ (<10^4^–<10^4^), <10^4^–3 × 10^7^ *n* (c > 10^4^) = 3 *n* (c > RV) = 3	<10^4^ (<10^4^–<10^4^), <10^4^–3 × 10^6^ *n* (c > 10^4^) = 3 *n* (c > RV) = 3	-
*Klebsiella oxytoca*	<10^4^	not detected	not detected	-
*Staphylococcus aureus*	<10^4^	<10^4^ (<10^4^–7 × 10^5^), <10^4^–5 × 10^7^ *n* (c > 10^4^) = 9 *n* (c > RV) = 9	<10^4^ (<10^4^–9 × 10^5^), <10^4^–8 × 10^6^ *n* (c > 10^4^) = 10 *n* (c > RV) = 10	-
*Clostridium difficile*	not detected	not detected	not detected	-
*Clostridium perfringens*	not detected	<10^5^ (<10^5^–<10^5^), <10^5^–1 × 10^7^ *n* (c > 10^5^) = 3 *n* (c > RV) = 3	<10^5^ (<10^5^–<10^5^), <10^5^–8 × 10^6^ *n* (c > 10^5^) = 4 *n* (c > RV) = 4	-
*Proteus vulgaris/mirabilis*	<10^4^	<10^5^ (<10^5^–<10^5^), <10^5^–2 × 10^9^ *n* (c > 10^5^) = 3 *n* (c > RV) = 3	<10^5^ (<10^5^–1 × 10^5^), <10^5^–3 × 10^7^ *n* (c > 10^5^) = 7 *n* (c > RV) = 7	-
*Enterobacter* spp.	<10^4^	2 × 10^6^ (<10^5^–3 × 10^7^), <10^5^–7 × 10^10^ *n* (c > 10^5^) = 17 *n* (c > RV) = 17	8 × 10^6^ (1 × 10^6^–5 × 10^7^), <10^5^–3 × 10^10^ *n* (c > 10^5^) = 22 *n* (c > RV) = 22	0.403
*Citrobacter* spp.	<10^4^	<10^5^ (<10^5^–<10^5^), <10^5^–5 × 10^13^ *n* (c > 10^5^) = 3 *n* (c > RV) = 3	<10^5^ (<10^5^–<10^5^), <10^5^–6 × 10^5^ *n* (c > 10^5^) = 2 *n* (c > RV) = 2	-
*Fusobacterium nucleatum*	not detected	<10^5^ (<10^5^–<10^5^), <10^5–^1 × 10^7^ *n* (c >10^5^) = 6 *n* (c > RV) = 6	<10^5^ (<10^5^–<10^5^), <10^5–^6 × 10^5^ *n* (c >10^5^) = 3 *n* (c > RV) = 3	-
*Parvimonas micra*	not detected	<10^5^ (<10^5^–<10^5^), <10^5^–2 × 10^6^ *n* (c > 10^5^) = 4 *n* (c > RV) = 4	<10^5^ (<10^5^–<10^5^), <10^5^–1 × 10^6^ *n* (c > 10^5^) = 4 *n* (c > RV) = 4	-
*Salmonella* spp.	not detected	not detected	not detected	-
*Shigella* spp.	not detected	not detected	not detected	-
*Bacteroides fragilis*/*Faecalibacterium prausnitzii* Ratio	0.01–100	106 (58–293) 13–40,000 *n* (c > RV) = 14	121 (63–250) 40–900 *n* (c > RV) = 17	0.387

**Table 6 jpm-13-00971-t006:** The levels of different parameters in vaccinated (*n* = 20) and unvaccinated patients (*n* = 19) with post-COVID-19 syndrome on the day of admission for rehabilitation, and the results of the Mann–Whitney U-test. *n* (c > RV) indicates the number of samples with a level higher than the corresponding reference value.

Parameter	Reference Values	Vaccinated Patients (*n* = 20)	Unvaccinated Patients (*n* = 19)	*p*-Value
Glucose, mmol/L	4.1–5.9	5.2 (4.8–5.4), 4.1–7.4 *n* (c > RV) = 2	5.7 (5.3–6.4), 4.9–10.5 *n* (c > RV) = 7	0.004
Alanine Transaminase (ALT), U/L	<50.0	16.2 (12.6–20.8), 9.2–43.4	21.6 (17.9–26.6), 10.0–44.5	0.030
*Bacteroides* spp.	10^9^–10^12^	2 × 10^13^ (7 × 10^12^–4 × 10^13^), 6 × 10^11^–4 × 10^15^ *n* (c > RV) = 18	4 × 10^13^ (1 × 10^13^–2 × 10^14^), 2 × 10^12^–8 × 10^14^ *n* (c > RV) =19	0.030
*Bacteroides fragilis/Faecalibacterium prausnitzii* Ratio	0.01–100	88 (33–191) 1–4000 *n* (c > RV) = 8	750 (111–1667) 43–400,000 *n* (c > RV) = 15	0.001

## Data Availability

Not applicable.

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
