# Peer review of "Promising Markers of Inflammatory and Gut Dysbiosis in Patients with Post-COVID-19 Syndrome"

_jpm, 2023, doi:10.3390/jpm13060971_

Round 1

Reviewer 1 Report

In this study, the author reveals the clinical, laboratory, and intestinal dysbiosis from major microbiota-associated microorganisms in patients with post-COVID-19 syndrome. It used a complete blood count, coagulation test, blood chemistry, biomarkers, and metabolites in serum samples, and a polymerase chain reaction (PCR) method with fluorescent detection for major microbiota-associated microorganisms, which were compared with the variables of healthy volunteers or reference ranges.

The manuscript can be accepted with "Minor revisions"…

 Although the authors described in the “Limitations” section that using real-time PCR to characterize the gut microbiota instead of 16s rRNA sequencing or whole genome sequencing limits the taxa that can be detected and does not allow the evaluation of minor bacterial species, it must be considered in the title. For example, the authors cannot say “microbiota disorders” because microbiota includes all sets of bacteria that live in the intestine in a symbiotic relationship of both commensal type and mutualism. The method used does not define this. I suggest changing the “microbiota disorder” to “intestinal dysbiosis”.

 In material and methods, Colonoflor-16 kits must be described thoroughly the bacteria and yeast identified by this method.  

 In the results and discussion, the discussion looks like the results… I suggest using statements in results that complement this section. In addition, intestinal dysbiosis must be associated with antibiotics treatment (clinical feature).

The discussion section cannot be sectioned and must be focused on relevant findings (significant statistical differences).

E.g.  

Hematology parameters:  Hemoglobin (p=0.027) and Hematocrit (p=0.027)

Coagulation test:  Prothrombin time (p=0.039)

Blood chemistry test: Lactate Dehydrogenase (p=0.046), Aspartate Transaminase (p=0.005), and Neuronspecific Enolase (p=0.038)

Metabolites: 4-Hydroxybenzoic (p=0.003)

Major microbiota-associated microorganisms: Bifidobacterium spp. (p=0.019) and Escherichia coli (p=0.023). Interestingly, Enterococcus spp. and Escherichia coli enteropathogenic are not detected in patients after 14 days. These findings should be associated with clinical features. 

Author Response

The authors express their deep gratitude to the Reviewer for agreeing to read our manuscript and for the proposed comments, which we tried to take into account as much as possible. Please, find our responses in attachments.

Reviewer 1

Although the authors described in the “Limitations” section that using real-time PCR to characterize the gut microbiota instead of 16s rRNA sequencing or whole genome sequencing limits the taxa that can be detected and does not allow the evaluation of minor bacterial species, it must be considered in the title. For example, the authors cannot say “microbiota disorders” because microbiota includes all sets of bacteria that live in the intestine in a symbiotic relationship of both commensal type and mutualism. The method used does not define this. I suggest changing the “microbiota disorder” to “intestinal dysbiosis”.

Dear Editor, thank you for your comment and suggestion, we accept it (line 2).

In material and methods, Colonoflor-16 kits must be described thoroughly the bacteria and yeast identified by this method.

We added the English version of the Operating Instructions in Supplementary 1.

In the results and discussion, the discussion looks like the results… I suggest using statements in results that complement this section. In addition, intestinal dysbiosis must be associated with antibiotics treatment (clinical feature).

We added the information about antimicrobial treatment in 9 from 39 patients during acute phase of COVID-19 based on medical records and described the detected intestinal dysbiosis in these 9 patients (lines 265-272):

“Of the 39 patients, 9 patients were treated with antibiotics during acute phase of COVID-19 based on medical records. They were cephalosporins – 44%, broad spectrum beta-lactamase penicillins – 33%, fluoroquinolones – 22%, macrolides – 11%, and carbapenems – 11%; in two patients broad spectrum beta-lactamase penicillins and cephalosporins, or carbapenems and fluoroquinolones were used. On admission, these 9 patients had a high level of total bacterial mass, in 2 patients Lactobacillus spp. were decreased; in 4 patients an increased level of Bacteroides fragilis/Faecalibacterium prausnitzii ratio (>100) was observed, and only in 3 patients Akkermansia muciniphila was detected.”

The discussion section cannot be sectioned and must be focused on relevant findings (significant statistical differences). E.g. Hematology parameters:  Hemoglobin (p=0.027) and Hematocrit (p=0.027), Coagulation test:  Prothrombin time (p=0.039), Blood chemistry test: Lactate Dehydrogenase (p=0.046), Aspartate Transaminase (p=0.005), and Neuronspecific Enolase (p=0.038), Metabolites: 4-Hydroxybenzoic (p=0.003)

The Discussion section no longer has sections and describe relevant findings.

Major microbiota-associated microorganisms: Bifidobacterium spp. (p=0.019) and Escherichia coli (p=0.023). Interestingly, Enterococcus spp. and Escherichia coli enteropathogenic are not detected in patients after 14 days. These findings should be associated with clinical features.

We added the information about these findings (lines 276-280):

“One of the two patients who had Escherichia coli enteropathogenic and the patient who had Enterococcus spp. at admission had a decrease in IL-6, 4-hydroxybenzoic and Bacteroides fragilis/Faecalibacterium prausnitzii ratio (<100) after 14 days of rehabilitation. In another patient, in whom Escherichia coli enteropathogenic was detected at admission, there was no positive dynamics in these indicators after rehabilitation.”

Also, we added the information about vaccination as Section 3.6

Reviewer 2 Report

The fundamental weakness of the study reported in the manuscript is the fact that the number of patients included in the study is by far too small. This limitation is even more relevant in view of the fact that the deviation from the reference ranges of those biomarkers whose values differ from the reference range is also very small. The authors must extend the study to larger cohorts.

Moreover, partly these deviations were in the range of the variation coefficients of the analytical methods applied.

Another fundamental weakness of the manuscript is that the quantitative deviations of biomarkers observed were not related to changes in the clinical condition of patients during rehabilitation. Thus, the observations remain without clinical correlate. The authors need to evaluate the biomarkers in relation to the clinical changes of the patients.

The quality of the English language of manuscript should be improved.

Author Response

The authors express their deep gratitude to the Reviewer for agreeing to read our manuscript. Please, find our responses in attachments.

Review 2

The fundamental weakness of the study reported in the manuscript is the fact that the number of patients included in the study is by far too small. This limitation is even more relevant in view of the fact that the deviation from the reference ranges of those biomarkers whose values differ from the reference range is also very small. The authors must extend the study to larger cohorts. Moreover, partly these deviations were in the range of the variation coefficients of the analytical methods applied.

We mentioned the information about small number of patients in the limitations (lines 471-472). However, we don’t understand the comment about “the deviation from the reference ranges of those biomarkers whose values differ from the reference range is also very small”. Our main markers of the post-COVID-19 syndrome included:

- 4-hydroxybenzoic acid – the metabolite which was not detected in healthy people at all and was detected at the level of 6.8 (5.2-8.6), 2.0-13.0, µmol/L in all patients at the admission. The limit of quantitation for this metabolite using gas chromatography-mass spectrometry is 0.5 µmol/L with the relative standard deviation less than 30%. Thus, the concentration of this metabolite was detected correctly and “the deviation from the reference ranges” is undeniably significant.

- succinic acid – the metabolite which was detected in healthy people at the range of 4.8 (4.4-6.0), 3.3-12.4, µmol/L and exceeded this range twice (14.0 (10.0-15.5), 8.0-25.0, µmol/L) in patients at the admission. The limit of quantitation for this metabolite using gas chromatography-mass spectrometry is 0.5 µmol/L with the relative standard deviation less than 30%. Thus, the concentration of this metabolite was detected correctly and “the deviation from the reference ranges” is undeniably significant.

- fumaric acid – the metabolite which was detected in healthy people at the range of 1.3 (1.1-1.5), 0.8-2.3, µmol/L and exceeded this range twice (2.4 (1.9-3.0),1.5-5.5, µmol/L) in patients at the admission. The limit of quantitation for this metabolite using gas chromatography-mass spectrometry is 0.5 µmol/L with the relative standard deviation less than 30%. Thus, the concentration of this metabolite was detected correctly and “the deviation from the reference ranges” is undeniably significant.

- interleukin-6 – the biomarker with the reference value of less than 7.0 pg/mL was detected higher than this value in 23 of 28 patients at admission with median value of 11.7 and IQR 8.1-15.5. The limit of quantitation for this biomarker using electrochemiluminescence on Cobas e411 analyzer is 1.5 pg/mL with the relative standard deviation less than 10%. Thus, the concentration of this biomarker was detected correctly and “the deviation from the reference ranges” is undeniably significant.

Another fundamental weakness of the manuscript is that the quantitative deviations of biomarkers observed were not related to changes in the clinical condition of patients during rehabilitation. Thus, the observations remain without clinical correlate. The authors need to evaluate the biomarkers in relation to the clinical changes of the patients.

The positive dynamics in the clinical changes of the patients was observed and mentioned in the manuscript (Section 3.1). Also, the absence in the positive dynamics in most laboratory parameters, which were out of reference values, was also thoroughly described in relevant sections and summarized in lines 307-313. Thus, the absence in the correlation of the biomarkers and the clinical changes of the patients is obvious.

The positive dynamics in the clinical changes of the patients was more likely due to the breathing exercises. Saturation was increased after 14 days of the rehabilitation and this fact was mentioned in the Section 3.1. and the positive dynamics in lung injury (%) was observed compared to acute COVID-19 (p < 0.001, the Wilcoxon signed-rank test):

“From 39 patients, included in our study, only 28 were treated for 14 days of rehabilitation. After the 14-day rehabilitation program patients (n = 28) were repeatedly examined by the therapist. They noted a decrease in shortness of breath after physical activity and an improvement in state of health. The patients’ oxygen saturation levels (SpO2) increased and ranged from 97 to 99%. In most patients, vesicular breathing was carried out in all parts of the lungs. Some patients (n = 20) agreed for CT scan examination after the rehabilitation (Table S1) and the positive dynamics in lung injury (%) was observed compared to acute COVID-19 (p < 0.001, the Wilcoxon signed-rank test).”

Reviewer 3 Report

Thank you for the opportunity to review this interesting manuscript. The authors describe different variables of patients with post-COVID-19 syndrome before and after a rehabilitation intervention. While their findings are of value for better description of the post COVID-19 disease and effect of such intervention, there are still some major issues that should be addressed.

1.      Methods:

a.      Study design: Were patients admitted for the sole purpose of the rehabilitation? Why patients were discharge before the end of the program?

b.      Study design: The inclusion and exclusion criteria refer to the initial acute COVID-19 disease or to their current condition when they initiated the rehabilitation? Did only patients with prior moderate COVID-19 were included or their post COVID-19 was of moderate severity? The chest CT foci was at their inclusion or during the acute COVID-19?

c.      Study design: What was your definition for post-COVID-19 syndrome? How did you determine its severity?

d.      Did you measure respiratory symptoms? Other respiratory variables (PFTs or else)?

e.      The process of the rehabilitation program and its included interventions should appear under the methods section.

2.      Results:

a.      PT was the only coagulation test with significant change after the intervention. However, its median and IQR values before and after intervention were practically similar. Please explain.

b.      The main aim of your intervention was "mostly aimed at restoration of the respiratory system using breathing exercises" (line 253). However, you did not report in your results about patients' symptoms and respiratory variables. Did you measure any of these variables? Was there a change in this area? Did they correlate with a change in the laboratory results?

3.      Discussion:

a.      "The latter proved difficult to treat" (line 261) – please explain why do you need to treat a lab result? Did these results correlate with symptoms or objective respiratory measurements?

b.      How did you come to the conclusion that rehabilitation program should be personalized for the different biomarkers? Did you try different rehabilitation methods? Once again, why did patients' baseline biomarkers are important? Why rehab should be evaluated based on lab results' levels?

c.      Patients with extra-pulmonary manifestations during the acute COVID-19 might signify a more inflammatory response, especially when more than one system is involved. This could have effect the authors results. I suggest the authors to add this subject to the discussion and use the following paper for that:

https://link.springer.com/article/10.1007/s00296-022-05106-3

d.      Did any of the included patients were vaccinated? COVID-19 vaccines are known to effect the disease process and inflammatory response and could alter patients results. I suggest the authors to add this subject to the discussion and use the following paper for that:

https://pubmed.ncbi.nlm.nih.gov/35536849/

e.      Limitations: The most significant limitations that were not mentioned by the authors are: 1. There is no control group of patients with post COVID-19 syndrome to show that the result of the different lab results are not due to time alone. 2. The change in gut-related variables could just be due to the change in patients' diet during rehabilitation and are not necessarily related to the COVID-19 disease or related to other disease outcomes.  

Minor grammar corrections are needed.

Author Response

The authors express their deep gratitude to the Reviewer for agreeing to read our manuscript and for the proposed comments, which we tried to take into account as much as possible. Please, find our responses in attachments.

Round 2

Reviewer 2 Report

The authors responded adequately to the reviewer's critical comments and were able to dispel doubts about the statistical reliability of the data. Although the reviewer's criticism of the low number of cases remains, in view of the interesting data on changes in metabolic biomarkers, acceptance of the manuscript is now recommended after minor editing of the English language.

The authors responded adequately to the reviewer's critical comments and were able to dispel doubts about the statistical reliability of the data. Although the reviewer's criticism of the low number of cases remains, in view of the interesting data on changes in metabolic biomarkers, acceptance of the manuscript is now recommended after minor editing of the English language.

Author Response

Dear Reviewer,

Thank you for taking the time to review our responses and the revised version of the manuscript. We are glad that we were able to successfully answer your questions, and we hope that the improved version of the manuscript will be understandable and interesting to the readers of the JPM. We tried our best in editing of the English language.

Reviewer 3 Report

I appreciate the opportunity to re-review this interesting manuscript.

The authors made significant changes and improved many parts of the manuscripts. The grammar has improved as well.  There are still some issues to be addressed, as following:

-        As I wrote in my previous review, you are not presenting the results of respiratory symptoms after your intervention, while it was the main aim of your trial. In the provided text, you only write: "They noted a decrease in shortness of breath after physical activity and an improvement in state of health" (lines 176-177). In my opinion, the authors have to describe how many patients improved, how did they measure symptomatic burden (any scales?), was it statistically significant, and if specific symptoms continue while others improved.

-        Similar to the last issue – where are the initial saturation results and their comparison with after the rehabilitation program? I wonder if an improvement of 1-2% saturation should count as significant, especially when you measure it by pulse oximeter.

-        I praise the authors for analyzing patients' baseline characteristics according to their vaccine status. These are very interesting results.

-        The authors found changes in PT, LDH, AST, NSE, 4-Hydroxybenzoic, Bifidobacterium spp., and Escherichia coli. It does not matter if they have return to normal ranges or not – they are changed after your intervention. Now the question should be - what is the relevance of these changes for the patient! If these changes do not correlate with symptoms, saturation, or CT results – why is it matter that they are changed?

The authors state in their response letter – "Thus, the absence in the correlation of the biomarkers and the clinical changes of the patients is obvious" – this is not obvious! Correlations between variables is not related with their change after an intervention. This is why statistics exist. You should examine for correlations between the significant biomarkers and change in respiratory variables.

-        Once again, I recommend the authors to use the following paper to ascertain their discussion: https://pubmed.ncbi.nlm.nih.gov/35536849/.

-        You cannot assume that gut microbiome or any other biomarker/test result is the cause for patients' respiratory condition! These are only associations and not causal interactions. Patients symptoms are from the COVID-19 disease and all you found are possible markers.   

Additional spelling mistakes: for example - "ago" (line 72). Please revise.

Author Response

Dear Reviewer,

Thank you again for your time and interest to our manuscript. We tried our best to answer your questions and apologize that we did not understand some of them from the first time. Also we asked a colleague to revise our English. Thus, we hope that revised version of our manuscript is more clear and usefull for readers of JPM.

Here is the answers:

The authors made significant changes and improved many parts of the manuscripts. The grammar has improved as well.  There are still some issues to be addressed, as following:

-        As I wrote in my previous review, you are not presenting the results of respiratory symptoms after your intervention, while it was the main aim of your trial. In the provided text, you only write: "They noted a decrease in shortness of breath after physical activity and an improvement in state of health" (lines 176-177). In my opinion, the authors have to describe how many patients improved, how did they measure symptomatic burden (any scales?), was it statistically significant, and if specific symptoms continue while others improved.

We added A questionary for COVID-19 survivors on admission as a new Supplementary 2 and this questionary was used only on the admission. That is the reason why the more detailed information was provided on the admission. On the day of discharge only general questions were addressed to the patients and this information was described in lines 176-177. Thus, we don’t have information to answer “how many patients improved, how did they measure symptomatic burden (any scales?), was it statistically significant, and if specific symptoms continue while others improved”. Moreover, we added this fact as a limitation in lines 504-505: “In addition, patients were not assessed for cognitive function and neurological dysfunction, and were not examined on the day of discharge with the same questionary (Supplementary 2) as on admission.”

-        Similar to the last issue – where are the initial saturation results and their comparison with after the rehabilitation program?

This information is described in Section 3.1 lines 171-172 “Oxygen saturation levels (SpO2) ranged from 95 to 96%” and 181-182 “The patients’ oxygen saturation levels (SpO2) increased and ranged from 97 to 99%.”

I wonder if an improvement of 1-2% saturation should count as significant, especially when you measure it by pulse oximeter.

According to the results of the Wilcoxon signed-rank test, the increase in the saturation was statistically significant (p<0.01) but we did not demonstrate this result as we understand that these changes could be within the pulse oximeter measurement error. However, we add this into the Section 3.1 lines 182-185: “According to the results of the Wilcoxon signed-rank test, the increase in the saturation during 14 days of the rehabilitation was statistically significant (p < 0.01) but these changes could be within the pulse oximeter measurement error (2% error in the range of oxygen saturation level 90-100%).”

-        I praise the authors for analyzing patients' baseline characteristics according to their vaccine status. These are very interesting results.

-        Once again, I recommend the authors to use the following paper to ascertain their discussion: https://pubmed.ncbi.nlm.nih.gov/35536849/.

We added the information about vaccine status of patients and the differences in some biomarkers in Section 3.6, and we thoroughly studied the paper https://pubmed.ncbi.nlm.nih.gov/35536849/, but we suppose that this paper in not suitable for our study because it describes the vaccination and severe COVID-19. However, we found the relevant studies about connection between vaccination and post-COVID-19 and added them into the limitations (lines 506-510):

“Despite the statistically significant differences in some variables in vaccinated and unvaccinated patients (Table 6), we cannot identify the role of vaccination in such differences, since our patients may have them before vaccination. However, there are studies describing the association of vaccination with post-COVID-19 symptoms [43,44], and this may be the subject for further research.”

-        The authors found changes in PT, LDH, AST, NSE, 4-Hydroxybenzoic, Bifidobacterium spp., and Escherichia coli. It does not matter if they have return to normal ranges or not – they are changed after your intervention. Now the question should be - what is the relevance of these changes for the patient! If these changes do not correlate with symptoms, saturation, or CT results – why is it matter that they are changed?

We suppose that there is no relevance of these changes for the patient because no correlation was found between lung injury and these variables, and we added this information into the Discussion Section (lines 326-332):

“The changes in hemoglobin. hematocrit, prothrombin time, lactate dehydrogenase, alanine transaminase, neuronspecific enolase, 4-hydroxybenzoic acid, Bifidobacterium spp. and Escherichia coli were statistically significant after the rehabilitation program (p<0.05). However, the absence of the correlation of these variables with the lung injury (Table S1) in some patients (n=20) who agreed for CT scan examination, indicates that these variables were not associated with the improvement in respiratory function of the patients after rehabilitation.”

The authors state in their response letter – "Thus, the absence in the correlation of the biomarkers and the clinical changes of the patients is obvious" – this is not obvious! Correlations between variables is not related with their change after an intervention. This is why statistics exist. You should examine for correlations between the significant biomarkers and change in respiratory variables.

As the clinical changes for all patients were accessed only using A questionary for COVID-19 survivors, which we added as a new Supplementary 2, and oxygen saturation, which was previously described as an unreliable method because of the measurement error, we decided to use the lung injury (%) of a number of patients (n=20), who agreed for CT scan examination after the 14-day rehabilitation, and calculate the correlation with all parameters from our study. The significant correlations at p<0.05 were revealed for the D-dimer and glucose, and we added this information into the Section 3.2 lines 211-213: “Moreover, in some patients (n = 20) who agreed for CT scan examination after the rehabilitation (Table S1), the positive correlation between lung injury (%) and D-dimer was observed – r=0.67.” and lines 228-230 “ In some patients (n = 20) who agreed for CT scan examination after the rehabilitation (Table S1), the positive correlation between lung injury (%) and glucose was observed – r=0.51.”

-        You cannot assume that gut microbiome or any other biomarker/test result is the cause for patients' respiratory condition! These are only associations and not causal interactions. Patients symptoms are from the COVID-19 disease and all you found are possible markers.  

We agree with the Reviewer that “all we found are possible markers” of the post-COVID-19 syndrome, which includes not only respiratory symptoms, but other “frequent symptoms are fatigue, headache, dyspnea, myalgia, hair loss and some other symptoms” (line 37-38). The initial cause of “Patients symptoms are from the COVID-19” and this is undoubted. But the reasons, which support the long-term post-COVID-19 syndrome, could be different, resulting into systemic disturbance, including the gut dysbiosis.